# A Multistage In Silico Study of Natural Potential Inhibitors Targeting SARS-CoV-2 Main Protease

**DOI:** 10.3390/ijms23158407

**Published:** 2022-07-29

**Authors:** Eslam B. Elkaeed, Ibrahim H. Eissa, Hazem Elkady, Ahmed Abdelalim, Ahmad M. Alqaisi, Aisha A. Alsfouk, Alaa Elwan, Ahmed M. Metwaly

**Affiliations:** 1Department of Pharmaceutical Sciences, College of Pharmacy, AlMaarefa University, Riyadh 13713, Saudi Arabia; 2Pharmaceutical Medicinal Chemistry & Drug Design Department, Faculty of Pharmacy (Boys), Al-Azhar University, Cairo 11884, Egypt; ibrahimeissa@azhar.edu.eg (I.H.E.); hazemelkady@azhar.edu.eg (H.E.); alaaelwan34@azhar.edu.eg (A.E.); 3Faculty of Pharmacy (Boys), Al-Azhar University, Cairo 11884, Egypt; ahmedabdelalim.edu@gmail.com; 4Department of Chemistry, University of Jordan, Amman 11942, Jordan; ahmadmqaisi98@gmail.com; 5Department of Pharmaceutical Sciences, College of Pharmacy, Princess Nourah bint Abdulrahman University, P.O. Box 84428, Riyadh 11671, Saudi Arabia; aaalsfouk@pnu.edu.sa; 6Pharmacognosy and Medicinal Plants Department, Faculty of Pharmacy (Boys), Al-Azhar University, Cairo 11884, Egypt; 7Biopharmaceutical Products Research Department, Genetic Engineering and Biotechnology Research Institute, City of Scientific Research and Technological Applications (SRTA-City), Alexandria 21934, Egypt

**Keywords:** SARS-CoV-2, main protease, structural similarity, pharmacophoric, docking, ADMET, DFT, MD simulations

## Abstract

Among a group of 310 natural antiviral natural metabolites, our team identified three compounds as the most potent natural inhibitors against the SARS-CoV-2 main protease (PDB ID: 5R84), M^pro^. The identified compounds are sattazolin and caprolactin A and B. A validated multistage in silico study was conducted using several techniques. First, the molecular structures of the selected metabolites were compared with that of **GWS**, the co-crystallized ligand of M^pro^, in a structural similarity study. The aim of this study was to determine the thirty most similar metabolites (10%) that may bind to the M^pro^ similar to **GWS**. Then, molecular docking against M^pro^ and pharmacophore studies led to the choice of five metabolites that exhibited good binding modes against the M^pro^ and good fit values against the generated pharmacophore model. Among them, three metabolites were chosen according to ADMET studies. The most promising M^pro^ inhibitor was determined by toxicity and DFT studies to be caprolactin A (**292**). Finally, molecular dynamics (MD) simulation studies were performed for caprolactin A to confirm the obtained results and understand the thermodynamic characteristics of the binding. It is hoped that the accomplished results could represent a positive step in the battle against COVID-19 through further in vitro and in vivo studies on the selected compounds.

## 1. Introduction

On 19 February 2022, the WHO reported that SARS-CoV-2 had infected 418,650,474 humans globally and deprived another 5,856,224 of their lives [1]. Humankind has previously suffered from coronaviruses, such as those that caused MERS-CoV and SARS-CoV in 2012 and 2003, respectively [2,3]. The dearth of available efficient treatments requires fast and precise movement in different research directions in order to find a cure. Computer-aided drug discovery is a fast and reliable research field that can be very useful in terms of determining the biological activity of a new drug minimizing effort, time, and costs [4]. Computer-aided drug discovery approaches can be categorized into two types: structure-based and ligand-based approaches. The starting point in the first (structure-based) approach is determination of the structure of the target enzyme or protein. The binding affinities and modes of different ligands (naturally isolated, synthesized, or hypothesized metabolites) against the identified protein are computed using molecular docking and/or molecular dynamic simulations programs. The second step in this approach is biological examination and optimization according to in silico results [5,6,7,8,9]. On the other hand, the second (ligand-based) approach starts with the ligand molecules, not the targeted enzyme. Usually, a known ligand is subjected to in silico studies using similarity models against several compounds to select the most similar compound(s). Examples of the second approach are QSAR, structural similarity, and pharmacophore modeling studies. This approach targets the enhancement of activity, as well as the discovery of new ligands [10].

Humans consistently rely on nature as a primary treatment of infections and diseases [11,12]. In the past, the major source of natural treatments was plants [13,14]. Recently, scientists have identified other sources, such as marine organisms and microbes [15,16]. Scientists have linked the biological activities of natural products to different types of secondary metabolites that are found in marine organisms and microbes, such as saponins [17,18], alkaloids [19], pyrones [20], steroids [21], isochromenes [22], flavonoids [23,24,25], diterpenes [26], and sesquiterpenes lactones [27,28,29].

Viral proteases were promising targets that led to the discovery of several approved treatments against dangerous resistant viruses, such as human immunodeficiency virus, by targeting the aspartyl protease and hepatitis C virus through targeting of the serine protease [30]. The function of M^pro^ within SARS-CoV-2 is to separate the two overlying large polyproteins (pp1a and pp1ab), leading to the activation of sixteen functional and non-structural proteins. The latter reaction is an indispensable step in viral replication. Consequently, its inhibition would lead to certain viral damage [31]. The main viral protease (Mpro) differs from human proteases in terms of both sequence and structure [32], which make Mpro a favorable target for drug design and discovery [33,34].

The M^pro^ enzyme is a homodimer protein that contains two protomers and comprises three domains (I, II, and III). The first and second domains are catalytic domains and are comprised of six antiparallel β-barrel structures. The third domain is the one that is responsible for enzyme dimerization and is composed of 5 α-helices. The M^pro^ forms a functional dimer via intermolecular interactions between the helical domains (Appendix A) [35,36]. The active site of M^pro^ is in the cleft between the first and second domains and is composed of four pockets (subsites): S1’, S1, S2, and S4 [37]. In silico studies are virtual methods that can predict the molecular properties of a given compound, as well as the molecular interactions of a particular protein. To confirm the obtained results, we applied several techniques. Additionally, we utilized a molecular dynamic simulation study to explore the compound–protein interaction for a given time to validate the acquired observations. However, we present our results as a validated study that saves time, effort, and costs and strongly suggests a potentially active metabolite against COVID-19. Additionally, we believe that these results should be followed by in vitro and in vivo studies. In response to the COVID-19 pandemic, scientists applied various in silico approaches to analyze SARS-CoV-2 structures [38], study potential natural inhibitors [39], introduce new drug targets [40], design and optimize the structures of peptide-mimetic inhibitors [41], design SARS-CoV-2 vaccines [42], and repurpose FDA-approved or previously known drugs [43]. Our team used various in silico approaches to discover a potential natural inhibitor. In silico screening of the potentialities of fifty-nine isoflavonoids against hACE2 and viral main protease suggested the superiority of four compounds [44]. Additionally, the anti-SARS-CoV-2 in silico potentialities of fifteen alkaloids [45], flavonoids [46], and two 2-phenoxychromone derivatives [47] were examined against five and eight crucial proteins, respectively. Recently, we utilized a multistep in silico technique to identify the most promising natural inhibitor among a large collection of metabolites against a specific COVID-19 enzyme. The most potent inhibitors against SARS-CoV-2 nsp10 were determined among more than 300 natural antiviral metabolites [48]. Furthermore, the most promising semisynthetic inhibitor against SARS-CoV-2 papain-like protease was selected among 69 candidates [49].

We selected a set of 310 secondary metabolites that are naturally originated and belong to various chemical classes, affording them diversity in terms of chemical structures. We depended on the literature by exploring various published papers, as well as review articles [50,51], that discussed antiviral natural products. We also employed special keywords to select diverse chemical classes, such as alkaloids [52], flavonoids [53], peptides [54], etc. All the selected compounds have previously exhibited antiviral activities. In this research, the discovery of a potential natural M^pro^ inhibitor, the mentioned group of the natural antiviral metabolites, was screened using several computational techniques (Figure 1).

## 2. Results and Discussion

### 2.1. Molecular Similarity

In silico molecular similarity is a tool that determines the degree of similarity between two or more molecules in terms of quantitative basics. In this study, Discovery Studio software was used to compute various physicochemical properties of the selected candidates depending on a Gaussian-type distance to demonstrate the degree of similarity between them and the ligand ((2-cyclohexyl-~(N)6-pyridin-3-yl-ethanamide (**GWS**)). The main idea of this study is that similarity in terms of chemical structure could be an essential key to determine similarity in binding with the targeted enzyme and therefore inhibition of its activity [55].

Herein, we report the examination of the molecular similarity of 310 reported natural antiviral metabolites (Appendix A) against the cocrystallized ligand (**GWS**) of M^pro^ using Discovery Studio software. The molecular properties that were studied in this work include the number of rotatable bonds (R-b), number of rings (Ri), aromatic rings (Ar-Ri) and hydrogen bonds that can be donated (HBD); hydrogen bonds that can be accepted (HBA); partition coefficient (ALog p); molecular weight (M. Wt); and molecular fractional polar surface area (MFPSA). All these properties were examined in both the examined metabolites and **GWS**. The metabolites were tested in six groups. Each group contains 50 members in ascending order, with the exception of the last group (which contains 60 metabolites). Each was separately subjected to a similarity check against **GWS** (Figure 1).

The results (Appendix A) establish that thirty metabolites exhibited similarities with the **GWS**. These metabolites were accordingly chosen for the pharmacophoric examination. Figure 2 describes the tested compounds with the highest similarity with **GWS**.

### 2.2. Docking Studies

In the current study, the thirty most similar candidates to **GWS** were subjected to molecular docking against M^pro^ (PDB ID: 5R84). The docking process was initially validated by redocking the cocrystallized ligand inside M^pro^. The RMSD value was 0.73 °A, affirming the utilized protocol’s validity (Appendix A).

The binding free energies (ΔG) of the tested ligands were determined in kcal/mol (Table 1), and orientations and binding interactions of the examined ligands were investigated. The binding poses with the best scores (modes and ΔGs) were chosen for further analysis. The output from MOE software was further visualized using Discovery Studio 4.0 software.

The binding interactions and orientation of **GWS** inside the active M^pro^ were analyzed. The results of *in-silico* protein-ligand interaction showed that the following amino acid in the protein target participates actively in the interactions with ligands: (GLN-189, MET-165, MET-49, HIS-41, ARG-188, SER-144, PHE-140, CYS-14, and LEU-141. HIS-163, ASN-142, and GLU-166).

The proposed binding mode of the co-crystallized ligand **GWS** revealed a ΔG of −21.39 kcal/mol. Three hydrophobic interactions were established between cyclohexyl moiety and HIS-41, MET-49, and MET-165 residues in the first pocket of M^pro^. Also, the amide linker moiety engaged in two hydrogen-bonding interactions with the essential amino acids ASN-142, and GLU-166. Furthermore, the pyridine ring interacted hydrophobically with GLU-166 and LUE-141 amino acid residues. Also, it formed a hydrogen bond with HIS163 in the second pocket of M^pro^ (Figure 3A and Appendix A).

The results of docking studies revealed that compounds **51**, **52**, **109**, **112**, **234**, **235**, **236**, **291**, **292**, **293**, **303**, and **305** have good binding score against M^pro^. Metabolites **112**, **291**, **292**, **293**, **303**, and **305** exhibited the highest energy scores with binding modes with good binding mode against the target enzyme. These compounds were selected to analyze their biding modes. Compounds **51**, **52**, **109**, **234**, **236**, and **235** failed to give good binding mode although their good binding score.

The binding modes of the metabolite **112** into the active site of M^pro^ were illustrated in Appendix A. Docking of metabolite **291** into the active pocket of M^pro^ revealed a ΔG of −18.13 kcal/mol. Such metabolite exhibited two hydrogen-bonding interactions: one with ASN-142, and another with GLU-166 amino acid in the liker region of M^pro^. Three hydrophobic interactions were computed between the terminal isopropyl tail and amino acid residues HIS-41, MET-49, and MET-165 in the first pocket of M^pro^. Indole moiety was directed into the second pocket and interacted hydrophobically with CYS-145 amino acid (Figure 3B and Appendix A).

Visualizing the binding mode of metabolite **292** indicated that such metabolite could tightly bind to the receptor with the highest ΔG of −18.48 kcal/mol. It formed three important hydrogen bonds with HIS-163, ASN-142, and GLY-143. In addition, the terminal long aliphatic tail was stabilized through hydrophobic interactions with HIS-41, MET-49, and MET-165 (Figure 3C and Appendix A).

The docking score of the metabolite **293** was −15.98 kcal/mol. Such metabolite occupied the active site of M^pro^ forming four hydrogen-bonding interactions with HIS-163, ASN-142, GLU-166, and PHE-140. As it is an isomer of metabolite **292**, the terminal tail of metabolite **293** was involved in hydrophobic interaction with HIS-41, MET-49, and MET-165 amino acids (Figure 3D and Appendix A).

The proposed binding mode of metabolite **303** revealed an energy score of −16.75 kcal/mol against M^pro^. In such a metabolite the nitrogen atom of tetrahydroquinoline ring formed only one hydrogen bond with the protein through interaction with ASN-142 amino acid. However, the phenyl ring of quinoline moiety reacted with the receptor through hydrophobic interaction MET-49. The carboxylic group formed electrostatic interaction with HIS-41. Furthermore, the terminal transeceoid aliphatic tail binds to the receptor by forming hydrophobic interactions with HIS-172 and HIS-163 (Appendix A).

The binding of metabolite **305** into the active site of M^pro^ resulted in an energy score of −17.31 kcal/mol. Metabolite **305** interacted with the protein via its hydroxyl and carbonyl groups forming two hydrogen bonds with ASN-142 and GLU-166 residues. Additionally, the phenyl ring formed hydrophobic interaction with MET-165 while the terminal isopropyl tail interacted with HIS-163 and HIS-172 amino acids via two hydrophobic interactions (Appendix A).

### 2.3. Pharmacophore Study

The word pharmacophore defines the key structural features in a compound to bind with a protein (enzyme) target resulting in the inhibition or the elicitation of a specific biological activity. The 3D-pharmacophoric model describes that feature in addition to their 3D geometry [56]. The optained3D model is an essential key that can be employed to expect a bioactivity of a compound according to the absence or presence of these features [57,58].

In the oulined study, we generated pharmacophore model from the biding pattern of **GWS** () with M^pro^. The generated model was validated internally by the used protocol. Then, such validated pharmacophore model was used as a further a confirmatory step for the docking output. The most similar thirty candidates produced from similarity check procedure were subjected to pharmacophore study to ensure that compounds have the main essential pharmacophoric features of **GWS**.

#### 2.3.1. Generation and Validation of the 3D-Pharmacophore Model

The pharmacophore model was generated using Discovery Studio 4.0 software. The cocrystallized ligan) of M^pro^ was used as a reference molecule. We studies how **GWS** binds with the active site in order to compute a 3D pharmacophore model. The software utilized several features in the process of pharmacophore generation, such as hydrogen bond donation (HBD), hydrogen bond acceptance (HBA), hydrophobic aliphatic groups (HA), hydrophobic aromatic groups (HAr), and ring aromatic (RA). Due to the lack of published data with respect to molecules that inhibited in vitro M^pro^ showing IC_50_ values, we could not run an external validation and depended on the internal validation that was performed the software to **GWS** inside the active pocket. The tested compounds were used as a training set.

The obtained 3D pharmacophore model represents three features: one H-bond donor besides two hydrophobic centers (Figure 4). It was employed as a 3D query to evaluate the tested metabolites as possible M^pro^ inhibitors.

#### 2.3.2. Test Set Activity Prediction

The thirty most similar candidates were mapped with the generated 3D pharmacophore model. As a result, the metabolites that verified the essential pharmacophoric features and the fit value were selected as candidates for the next step.

The results privileged twelve metabolites that had the main essential features of M^pro^ inhibitors. Surprisingly, some metabolites showed higher fit values than **GWS** against and the generated 3D pharmacophore. In particular, metabolites **291** (fit value = 2.587), **292** (fit value = 2.887), **293** (fit value = 2.890), **303** (fit value = 2.867), and **305** (fit value = 2.792) showed high fit values compared to the cocrystallized ligand (fit value = 2.804) (Table 2). Figure 5 and Figure 6 show the mapping of the twelve best metabolites against the generated 3D pharmacophore.

According to the pharmacophoric studies, all the chosen metabolites were reported to be strong antivirals. Individually, indican (**15**) is a colorless glucoside isolated from *Indigofera suffruticosa* [59] and *Isatis indigotica* [60]. Interestingly, indican inhibited the cell-free cleavage activity of the SARS-CoV 3CL^pro^, with an IC_50_ value of 112 μM [61]. The beta-carboline alkaloid harmine (**30**), which was identified for the first time in 1847 from seeds of *Peganum harmal* [62] interacted in silico against M^pro^, with a binding affinity value of −6.3 kcal/mol [63]. The famous furanochromone derivative khellin (**55**) of *Ammi visnaga* [64] blocked the viral gene expression of murine cytomegalovirus in vitro through a photo-induced mechanism [65]. Sessiliflorol A (**109**) and syzygiol (**112**) are phloroglucinol derivatives that were reported in *Melicope sessiliflora* [66] and *Syzygium polycephaloides* [67], respectively. Sessiliflorol A exhibited in vitro antiviral properties against both types of herpes simplex virus I and II, with IC_50_ values of 22.3 and 10.4 μM, respectively [66], whereas syzygiol (**112**) inhibited the activation of Epstein–Barr virus [68]. Spongiadiol (**234**) and isospongiadiol (**236**) are diterpene derivatives identified from a marine sponge of the *Spongia* species. Metabolites **234** and **236** exhibited activities against herpes simplex virus type 1 (HSV-1), with IC_50_ values of 0.25 and 2 μg/mL, respectively, compared to the standard, acyclovir, 0.5 μg/mL [69].

The potent antiviral acyloin sattazolin (**291**) was isolated from a soil bacterium of *Bacillus* sp. and exhibited activities against HSV-1 and HSV-2, with an ID_50_ of 1.5 µg/mL [70]. The two caprolactams caprolactin A and B (**292** and **293**) were isolated from an unidentified bacterium of a sample collected from deep-ocean sediment. Caprolactin A and B exhibited antiviral activity towards HSV-2 at a concentration of 100 μg/mL [71]. Virantmycin (**303**) is a chlorine-containing metabolite that was isolated from *Streptomyces nitrosporeus* and inhibited eight RNA and DNA viruses (vesicular stomatitis virus (VSV), Sindbis virus (SbV), Western equine encephalitis virus (WEE), Newcastle disease virus (NDV), vaccinia virus (Vac-DIE, DIE and IHD strains), HSV-1, and HSV-2, with EC_50_ values of 0.008, 0.006, 0.003, 0.04, 0.005, 0.004, 0.03, and 0.02 µg/mL, respectively [72,73]. Sattabacin (**305**) was isolated from a bacterium belonging to *Bacillus* sp. and inhibited HSV-1 and HSV-2, with an ID_50_ of 3 µg/mL, showing selective inhibition against protein synthesis in the infected cells [70].

### 2.4. ADMET Studies

Five metabolites—**291**, **292**, **293**, **303**, and **305**—exhibited the highest binding affinity against M^pro^. In addition, these five compounds showed the highest fit values against the pharmacophore mode. These metabolites were further investigated for their pharmacokinetic properties (ADMET studies) using Discovery Studio 4.0. Indinavir is an effective antiviral drug targeting viral protease, and it was used as a reference molecule in this study.

The results revealed that metabolites **291**, **292**, and **293** exhibited the same acceptable values of ADMET parameters (Table 3). These metabolites have a medium level of blood–brain barrier (BBB) penetration, a good level of aqueous solubility, and a good absorption level from the human intestine and are non-inhibitors of CYP2D6, with plasma protein-binding ability of less than 90%. These values indicate that these metabolites have an acceptable range of drug likeness. Metabolite **303** was predicted to be a CYP2D6 inhibitor. Therefore, hepatotoxicity may be expected upon administration of this metabolite. Metabolites **303** and **305** were expected to bind the plasma protein by more than 90% leading to decreased distribution and bioavailability. Furthermore, metabolite **305** was predicted to exhibit a high level of BBB penetration, which may produce central nervous system (CNS) side effects. Figure 7 demonstrates the examined ADMET profiles represented as ellipses: lipid–water partition coefficient (AlogP98, blue point), intestinal absorption (95% confidence limit (red ellipse) and 99% confidence limit (green ellipse)), and blood–brain barrier (BBB) 95% confidence limit (pink ellipse) and 99% confidence limit (turquoise ellipse)). As presented in Figure 7, the five points lie in the area encompassed by the four ellipses, indicating that compounds **291**, **292**, **293**, **303**, and **GWS** have good absorption levels and medium BBB penetration levels. The point lies outside the pink and turquoise ellipses and inside the red and green ellipses, indicating that compound **305** has a high BBB penetration level and a good absorption level.

### 2.5. Toxicity Studies

The toxicity profiles of metabolites **291**, **292**, and **293** were tested against eight different models using Discovery Studio software [74,75]. Indinavir was used as a reference drug.

In general, metabolites **292** and **293** showed lower toxicity potential than metabolite **291**. In particular, metabolite **291** was predicted to be non-carcinogenic against the FDA rodent carcinogenicity model (FDA-RC) and showed carcinogenic potency in mice (M-TD_50_), with a value of 136.851 mg/kg/day, which is higher than that of indinavir (6.954). (The maximum tolerated dose in rats (MTD-R) of metabolite **291** was 0.346 g/kg, which is higher than that of indinavir (0.133). In terms of developmental toxicity potential, metabolite **291** was predicted to be toxic. Metabolite **291** had a rat chronic LOAEL of 0.081 g/kg, which is higher than that of indinavir (0.008). Finally, metabolite **291** was predicted to be an irritant drug in both ocular and skin irritancy models.

Metabolites **292** and **293** were predicted to be non-carcinogenic against the FDA rodent carcinogenicity model. They showed carcinogenic potency, with TD_50_ values of 104.247 and 108.043 mg/kg/day, respectively, which are both higher than that of indinavir. These metabolites showed an equal rat maximum tolerated dose of 0.126 and g/kg, which is almost equal to that of indinavir. In terms of developmental toxicity potential, metabolites **292** and **293** were predicted to be non-toxic. Metabolites **292** and **293** had rat chronic LOAEL values of 0.541 and 0.456 g/kg, respectively. These values are far higher than that of indinavir. Finally, such metabolites were predicted to be non-irritant against both ocular and skin irritancy models. Accordingly, metabolites **292** and **293** were advantaged for further investigations (Table 4).

### 2.6. DFT Studies

DFT parameters, including total energy [76], HOMO [77], LUMO [77], gap energy [78], and dipole moment [79,80], were calculated for metabolites **292** and **293** by Discovery Studio software using **GWS** as a reference. The function used in this test was PWC of local density approximation (LDA). Also, the quality was selected to be coarse using a DN basis set with an SCF density convergence of 1.0 × 10^−4^ with Accelrys in the DMol3 module of the Materials Studio package 

#### 2.6.1. Molecular Orbital Analysis

As shown in Table 5, metabolites **292** and **293** had dipole moment values of 2.084 and 2.100, respectively. These values are higher than that of **GWS** (1.708). The elevated dipole moment is expected to increase hydrogen bonding, as well as non-bonded interactions in the metabolite–protein complexes that are predicted to increase the binding affinity during SARS-CoV-2 inhibition.

Additionally, HOMO energy indicates the region of the examined compond, which can be an electron doner during rbinding. On the other hand, LUMO energy identifies the region that can work as an electron acceptors. HOMO-LUMO gap energy is the difference between HOMO and LUMO energies, represents the electronic excitation energy that is essential to calculate the molecular reactivity as well as stability of the examined compound.

Furthermore, metabolites **292** and **293** had equal gap energy values of 0.183 Ha, which are higher than that of **GWS** (0.129 Ha). The increased gap energy of metabolites **292** and **293** indicates a the increased stability of these compounds. Figure 8 shows the spatial distribution of molecular orbitals for metabolites **292**, **293**, and **GWS**.

#### 2.6.2. Electrostatic Potential Map

Calculation of the electrostatic interactions can be used to evaluate the energy of the M^pro^ metabolite complexes [81]. Electrostatics are a main forces involved in the process of molecular recognition [82].

On the MEP surface (Figure 9), every electronegative atom shows a negative value of charge (represented in red; H-bonding acceptor). On the other hand, the electron-poor atom shows a positive value (represented in blue; H-bonding donor). The atoms with zero values (represented in green to yellow) are neutral [83].

Valuable insight can be gained about the binding pattern with the receptor from molecular orbital and electrostatic potential map analyses (Figure 8 and Figure 9). The HOMO of **GWS** was concentrated on a pyridine moiety and the C=O group of the amide linker (two red patches). These functional groups formed two hydrogen-bond acceptors with His163 and Glu166. The LUMO of **GWS** was concentrated in the NH group of amide the linker (blue patch), which formed a hydrogen bond donor with Asn142. In addition, the pyridine and cyclohexyl rings have a high electron density (green to yellow), which can favor the hydrophobic and π-staking interactions withLeu141, Glu166, Met49, Arg188, and His41.For compound **292**, the HOMO was concentrated on the two C=O groups (two red patches), forming two hydrogen-bond acceptors with Gly143 and His163, whereas the LUMO was concentrated on the two NH groups (two blue patches), forming two hydrogen-bond donors with His172 and Glu166. Regarding compound **293**, the HOMO was concentrated on the two C=O groups (two red patches), forming two hydrogen-bond acceptors with Glu166 and His163, whereas the LUMO was concentrated on the two NH groups (two blue patches), forming three hydrogen-bond donors with Phe140, Asn142, and Glu166. The aliphatic chains in each molecule showed yellow patches that can form hydrophobic interactions with hydrophobic amino acid residues (Figure 9). Subsequently, metabolites **292** and **293** were identified as having the same probabilities in DFT studies.

Although metabolites **292** and **293** had the same probabilities in the DFT studies, **292** was selected for the next study, as it had higher binding energy in the docking studies.

### 2.7. Molecular Dynamic Simulation

The structural stability of metabolite **292** controlled depending on the calculation of the root-mean-square deviation (RMSD) over the M^pro^ backbone atoms, in addition to the root-mean-square fluctuation (RMSF). Furthermore, the hydrogen bonding were analyzed over a 150 ns NPT ensemble.

#### 2.7.1. Trajectory Analysis

The stability of the caprolactin A-M^pro^ complex was verified by the RMSD of backbone atoms between the initial modeled structure and the simulated structure over 150 ns. As expected, an increase in RMSD compared to the modeled structure was observed after 10–20 ns, and no significant development of RMSD was observed until 50 ns. The simulations converged between 2 and 5 Å (Figure 10A). To assess the effect of M^pro^ interaction on these residues, we compared the theoretical B-factor data using RMSF plots. Relative stabilizations were observed for the M^pro^ residues that are responsible for formation of the caprolactin A-M^pro^ complex (Figure 10B). Moreover, M^pro^ reached a stable conformation, with the radius of gyration fluctuating around 22.7 Å (Figure 10C).

#### 2.7.2. Binding Analysis: Hydrogen Bonds and Contact Frequency

The hydrogen bonding occupancy of caprolactin A-M^pro^ complex was calculated as the fraction of conformations out of 1500 conformations in which caprolactin A participates. The 1500 conformations were obtained from the corresponding 150 ns molecular dynamics trajectory. The H-bond occupancy table for caprolactin A is shown in Table 6.

The average number of H-bonds between caprolactin A-M^pro^ complex was ~1, with (0) as a minimum value and (2) as a maximum value (Appendix A).

Furthermore, caprolactin A binding was stabilized by a network of hydrogen bonds connecting the key amino acid residues His41 and Glu166 to NH and OH of the molecule, respectively. Hydrophobic interactions involving the amino acid residues Met49, Met165, and Cys145 also contributed to the binding interaction (Appendix A).

To estimate the binding between the caprolactin A-M^pro^ complex, a contact frequency (CF) analysis was performed utilizing the *contactFreq.tcl* module on VMD and with a cutoff of 4 Å. Table 7 represents the results of the experiment. In the simulation study, the following amino acid residues exhibited higher CF values: Thr25, His41, Met49, Asn142, Cys145, His164, Met165, Glu166, Asn187, and Gln189 (Appendix A).

#### 2.7.3. Calculation of Free Energy by MM/(GB)SA

To calculate the relative binding energy, the “one-average molecular mechanics generalized Born surface area (MM/GBSA) approach” [84,85] was utilized. Caprolactin A exhibited an MMGBSA relative binding free energy against M^pro^, with a value of −76.92 kJ mol^−1^.

The preceded molecular dynamics (MD) simulations studies explained the thermodynamic characteristics of metabolite **292** and provided deep insights with respect to its binding against the M^pro^. The obtained results confirmed our choice of caprolactin A (**292**). Caprolactin A (**292**) is natural caprolactam that exhibited promising activity against HSV-2 at a concentration of 100 μg/mL [71]. Although **292** was isolated in the form of an inseparable mixture with its structural isomer caprolactin A, it was successfully and simply synthesized [71]. The reported total synthesis of metabolite **292** opens the door to synthesize this promising metabolite in larger amounts to conduct further in vitro and in vivo investigations.

## 3. Method

### 3.1. Molecular Similarity Detection

Molecular similarity of the 310 natural compounds against **GWS** was determined using Discovery Studio 4.0., 2016 (Vélizy-Villacoublay, France). The protocol was adjusted to determine the most similar 10%. The default molecular properties were applied [86,87,88]. More details are available in the Appendix A.

### 3.2. Pharmacophoric Study

The pharmacophore model applied using Discovery Studio 4.0 software. The protocol of receptor–ligand pharmacophore generation was applied using **GWS** as a reference molecule [89,90,91]. More details are provided in the Appendix A.

### 3.3. Docking Studies

The docking investigation was accomplished using MOE2014 and Discovery Studio 4.0 software [92,93,94,95,96]. More details are provided in the Appendix A.

### 3.4. ADMET Analysis

ADMET descriptors of the compounds were determined using Discovery Studio 4.0 [97,98,99,100]. More details are provided in the Appendix A.

### 3.5. Toxicity Studies

The toxicity parameters of the tested compounds were calculated using Discovery Studio 4.0. Indinavir was used as a reference drug [101,102,103]. More details are provided in the Appendix A.

### 3.6. DFT Studies

The examined compounds were subjected to the DFT calculation protocol using the default option [104,105]. More details are provided in the Appendix A.

### 3.7. Molecular Dynamic Simulations

Among the tested compounds, the most promising metabolite (**292**) was advanced to MD simulations to study the relative stability of the protein–ligand interactions. All simulations were performed using the NAMD 2.13 package and the CHARMM36 force field [106,107,108,109,110,111,112,113,114,115]. More details are provided in the Appendix A.

## 4. Conclusions

Three metabolites (sattazolin and caprolactin A and B) were identified as the most potent natural M^pro^ inhibitors among a group of 310 reported natural antiviral metabolites. Several computational techniques were applied in the presented study. The structural similarity detection against the co-crystallized ligand of M^pro^ (PDB ID: 5R84) selected the 30 most similar metabolites. Docking and pharmacophore studies favored five metabolites exhibiting correct binding modes against M^pro^ (PDB ID: 5R84) and good fit values against the generated pharmacophore model. Subsequently, ADMET, toxicity, and DFT studies were carried out to select the most promising inhibitor as caprolactin A. The molecular dynamic simulation studies confirmed the effective binding of caprolactin A with M^pro^ over 150 ns. Further studies should be carried out for such a promising metabolite that could represent hope for humankind in the fight against the COVID-19 pandemic.

## Data Availability

Data are available with the corresponding authors upon request.

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
