# Peer review of "A Multistage In Silico Study of Natural Potential Inhibitors Targeting SARS-CoV-2 Main Protease"

_ijms, 2022, doi:10.3390/ijms23158407_

Round 1

Reviewer 1 Report

In attention of the manuscript authors,

In the “ijms-1811462” manuscript, the author has made substantial research efforts to identify  the most promising natural inhibitor among a large collection of metabolites against a specific COVID-19 enzyme. To reach their goal, an in silico approach involving structural similarity search, pharmacophore, molecular docking ADMETox, DFT parameters computation, and molecular dynamic simulations were applied. As a final result, 1 out of 310 natural antiviral natural metabolites, caprolactin A, was selected as a promising SARS-Cov-2 inhibitor. 

The methodology and results if reconfigured, and presented in a more convincing way, will substantially improve the manuscript quality and could be a real gain for researchers interested in developing promising new natural antiviral natural metabolites candidates to combat COVID-19.

Considering the potential impact of the manuscript outcomes in the research world, I recommend that the manuscript be accepted for publication in Int. J. Mol. Sci. journal following major revision

Some of the manuscript shortcomings:

  1. The manuscript English must be revised
  2. The manuscript aspect must be improved. There are too many figures, and tables and the information in them is repeated. The quality of figures must be improved (e.g. Figs. 4,5,6, etc)
  3. What about the selection of 310 natural metabolites. How were they selected?
  4. The authors talk about a large collection of metabolites. Which one?
  5. The authors made structural similarity searches and selected 30 compounds as the most similar. Please keep only the structure of the compounds and move to supplementary Figure 1 and Table 1. The latter describes the same information.
  6. I do not see the role and relevance of the pharmacophore. The pharmacophore must be applied to a collection of compounds in order to find new candidates with the same or improved features, in no case apply it to the same compounds.
  7. The normal way is to dock the 30 selected compounds into the target active site and select the best candidates based on the docking score and appropriate interactions. Moreover, please keep in the manuscript only the 2D version of the interaction. It is easier for the reader to see the results.
  8. The discussion about the redocking step of the co-crystallized ligand is too long. Please be short and keep the relevant aspects (RMSD, docking score, interaction).moreover, the figure clearly describes the quality of the docking preparation step.
  9. Why compound 55 was not discussed even though it has a docking score close to 293.
  10. Also, the docking paragraph must be reconfigured from figures (try to combine them and keep only 2D) and information points of view.
  11. Why does Indinavir appear as a reference only in the ADMETox steps? it was interesting and maybe even normal to be considered in docking.
  12. The authors present that an “increased gap energy of metabolites 292 and 293 indicates the higher possibility to interact with the target receptor”. It is not very clear how the gap energy indicated that thing?
  13. What is the main utility of DFT studies in the selection of the best candidates? This study supports only a few docking observations.
  14. In the opinion of the referee, all 3 compounds should have been evaluated.

Author Response

Some of the manuscript shortcomings:

1- The manuscript English must be revised

Response: Thank you for your efforts and your valuable comments. The whole manuscript was revised

2- The manuscript aspect must be improved. There are too many figures, and tables and the information in them is repeated. The quality of figures must be improved (e.g. Figs. 4,5,6, etc).

Response: Done

3- What about the selection of 310 natural metabolites. How were they selected?

Response: Reason has been justified (We selected a set of 310 secondary metabolites that share being naturally originated and belong to various chemical classes which gives them diversity in chemical structures. All the selected compounds have exhibited antiviral activities before.)

4- The authors talk about a large collection of metabolites. Which one?

Response: we mean the selected 310 secondary metabolites

5- The authors made structural similarity searches and selected 30 compounds as the most similar. Please keep only the structure of the compounds and move to supplementary Figure 1 and Table 1. The latter describes the same information.

Response: DONE

6- I do not see the role and relevance of the pharmacophore. The pharmacophore must be applied to a collection of compounds in order to find new candidates with the same or improved features, in no case apply it to the same compounds.

Response: there are many ways to generate pharmacophore. 1) from bioactive conformation depending one active compound. 2) from receptor ligand complex: in this case you can generate the pharmacophore model from the binding mode of the co-crystallized ligand with the target protein. 3) from a set ligand: you can generate the pharmacophore mode using a training set of active and in active molecules.

In our manuscript, we depended on the second one and we generated the pharmacophore form the binding interaction of the co-crystallized ligand in the active site. In this approach, the generated pharmacophore is internally validated, and the invalid model would be excluded.

7- The normal way is to dock the 30 selected compounds into the target active site and select the best candidates based on the docking score and appropriate interactions. Moreover, please keep in the manuscript only the 2D version of the interaction. It is easier for the reader to see the results.

Response:

The 3D figures were transferred into the supplementary data as requested.

We used the pharmacophore as a filtration step which gave 12 compounds subjected for the docking studies. If it is necessary to carry out the docking studies for 30 compounds, we can delete the pharmacophore study and perform that. Please, let me know what is appropriate.

8- The discussion about the redocking step of the co-crystallized ligand is too long. Please be short and keep the relevant aspects (RMSD, docking score, interaction). moreover, the figure clearly describes the quality of the docking preparation step.

Response: DONE

9- Why compound 55 was not discussed even though it has a docking score close to 293.

Response: Although it showed good score, but its binding mode was not similar to the co-crystallized ligand.

10- Also, the docking paragraph must be reconfigured from figures (try to combine them and keep only 2D) and information points of view.

Response: we transferred the 3D figures and kept only the 2D ones.

11- Why does Indinavir appear as a reference only in the ADMETox steps? it was interesting and maybe even normal to be considered in docking.

Response: we used Indinavir as a reference molecule in ADMET and toxicity studies because this compound is an FDA approved and it was logic to compare the kinetic and toxicity with approved compound. While in the docking studies, we used the co-crystallized ligand as a reference since it is the highest efficient molecule to bind the active site. In addition, our rationale was based on the detection of similarity with the co-crystallized ligand, and it was necessary to use this ligand as a reference in the docking studies to reach a good comparison in the binding modes against the target protein.

12- The authors present that an “increased gap energy of metabolites 292 and 293 indicates the higher possibility to interact with the target receptor”. It is not very clear how the gap energy indicated that thing?

Response: thank you for this notice. This sentence was corrected.

13- What is the main utility of DFT studies in the selection of the best candidates? This study supports only a few docking observations.

Response: you are right. We tried to use the DFT as a filter to select the most efficient molecule against the active site but the result of the DFT did not give us additional filtration. However, it confirmed the results of the docking studies. Accordingly, it is considered a confirmatory test in this study.

14- In the opinion of the referee, all 3 compounds should have been evaluated.

Response: Unfortunately, we don’t have access to in vitro COVID-19 assay at the right time.

However, the publishing of this work will give the opportunity for all scientists in the world to examine these interesting compounds and it will be an important step in the battle against COVID-19

Reviewer 2 Report

Dear authors,

A very big amount of in silico work and effort, which unfortunately is not supported by used syntax and grammar.

In my opinion, extended English language editing throughout the text is needed in order for the readers to get the meaning. Articles are either used inappropriately or excessively. Past tense is messing up the overall text meaning in many cases. Additional letters are found in several words.

Moreover, conclusions section should be enriched to provide useful information extracted by the methods.

Author Response

In my opinion, extended English language editing throughout the text is needed in order for the readers to get the meaning. Articles are either used inappropriately or excessively. Past tense is messing up the overall text meaning in many cases. Additional letters are found in several words.

Response: Thank you for your efforts and your valuable comments. The manuscript has been extensively revised and all changes were highlighted in the revised manuscript.

Moreover, conclusions section should be enriched to provide useful information extracted by the methods.

Response: conclusion has been modified

Reviewer 3 Report

Metwaly, Elkaeed, and co-workers present an in silico study to determine the inhibitory potential of a group of 310 natural products on the SARS-CoV-2 main protease (Mpro) enzyme. A filtration protocol based on different computational techniques is employed to validate the druggability of the selected metabolites. The theoretical data reveals that caprolactin A has the best thermodynamic features and the adequate ADMET profile to be considered a promising Mpro inhibitor.

Although the proposed computational protocol accurately examines most of the key aspects that any ligand should comply with to be considered a potential drug, this study presents some deficiencies that must be corrected in order to be acceptable for publication in the International Journal of Molecular Sciences:

1) In the introduction section there is no detailed discussion of previous computational studies performed by other authors based on the inhibitory effect of small molecules on SARS-CoV-2 enzymes. Summarizing briefly the results and computational strategy followed by different authors may help to contrast and validate the proposed protocol by Metwaly et al.

2) There is no justification for the chosen 310 natural compounds. Why are these compounds selected and not other natural products?

3) It is not indicated in the main text which physicochemical properties underly the molecular similarity calculation. I suggest moving the description of these properties already present in the Supporting Information (SI) to the "molecular similarity" section. It lacks explaining in more detail why the 30 selected compounds are similar to the GWS according to these properties.

4) The x-y-z coordinates of the active site of the Mpro enzyme are not included in the SI. It should be desirable to include this information in order to give the opportunity to reproduce the results.

5) The significance of Figure 14 is not properly accounted for. What kind of information gives us?

6) In the DFT study section, the total energy of every ligand has nothing to do with binding to the target receptor. This energy depends on the number of atoms present in the ligand (the more atoms, the more negative the energy). In table 6, how is calculated the binding energy? Why does increased gap energy indicate a higher possibility to interact with the target receptor? It has no sense to include the molecular orbitals of the metabolites if they are not put in relation to their ability to interact with Mpro. No information is given about the employed DFT method (functional, basis set, etc.)

7) It would be interesting to correlate the conclusions extracted from the MEP surfaces with the interactions with residues observed in the docking studies. This will clarify the role of every fragment of the ligands in protein-ligand complexes.

8) Change "anti-SARA-CoV-2" in line 93 to "anti-SARS-CoV-2" and define the expression "CNS" in ADMET section.

The previous issues should be addressed and solved before publication.

Author Response

Thank you for your praise and valuable comments. We considered these comments with high interest. The comments and our responses are summarized in the following points.

1- In the introduction section there is no detailed discussion of previous computational studies performed by other authors based on the inhibitory effect of small molecules on SARS-CoV-2 enzymes. Summarizing briefly the results and computational strategy followed by different authors may help to contrast and validate the proposed protocol by Metwaly et al.

Response: Done

2- There is no justification for the chosen 310 natural compounds. Why are these compounds selected and not other natural products?

Response: Reason has been justified (We selected a set of 310 secondary metabolites that share being naturally originated and belong to various chemical classes which gives them diversity in chemical structures. All the selected compounds have exhibited antiviral activities before.)

3- It is not indicated in the main text which physicochemical properties underly the molecular similarity calculation. I suggest moving the description of these properties already present in the Supporting Information (SI) to the "molecular similarity" section. It lacks explaining in more detail why the 30 selected compounds are similar to the GWS according to these properties.

Response: have been indicated

4- The x-y-z coordinates of the active site of the Mpro enzyme are not included in the SI. It should be desirable to include this information in order to give the opportunity to reproduce the results.

Response: The x-y-z coordinates of the active site of the Mpro enzyme are now included in the SI.

5- The significance of Figure 14 is not properly accounted for. What kind of information gives us?

Response: additional clarifications were added to this figure in the revised manuscript.

6- In the DFT study section, the total energy of every ligand has nothing to do with binding to the target receptor. This energy depends on the number of atoms present in the ligand (the more atoms, the more negative the energy). In table 6, how is calculated the binding energy? Why does increased gap energy indicate a higher possibility to interact with the target receptor? It has no sense to include the molecular orbitals of the metabolites if they are not put in relation to their ability to interact with Mpro. No information is given about the employed DFT method (functional, basis set, etc.).

Response:  

  • You are right regarding the total energy. So, we deleted that from the discussion due to the non-significant value.
  • For the calculated binding energy in table 6, refers to the interaction energy between all the atoms in the molecule. Also known as cohesive energy.
  • For the gap energy, the discussion was corrected. The increased gap energy of metabolites 292 and 293 indicates the higher stability of these compounds.
  • The relation between molecular orbitals of the metabolites and their ability to interact with Mpro was clarified in the revised manuscript.
  • The information about the employed DFT method was added in the revised manuscript.
  • 7- It would be interesting to correlate the conclusions extracted from the MEP surfaces with the interactions with residues observed in the docking studies. This will clarify the role of every fragment of the ligands in protein-ligand complexes.

Response: done

8- Change "anti-SARA-CoV-2" in line 93 to "anti-SARS-CoV-2" and define the expression "CNS" in ADMET section.

Response: has been changed

Round 2

Reviewer 1 Report

In attention of the manuscript authors,

The authors responded satisfactorily to all referee’s requirements and made all the changes addressed in the manuscript. The manuscript has been substantially improved in both chemical content and English but some improvements are still required. Therefore, if the referee’s suggestions are applied, the manuscript will meet the journal requirements and should be considered for publication.

In this context, I agree to recommend that the manuscript be accepted for publication in the IJMS journal following major revision.

Please find enclosed some of the manuscript shortcomings:

11. Figures 6 and 7 should be merged into one figure, although, in the referee's opinion, both figures could be moved to supplementary and keep only a brief explanation.

22. The authors added a short explanation about the selection of the 310 secondary metabolites. The explanation is not satisfactory. Where did the authors find these compounds? In literature or any other resources? If literature, please add the appropriate references. If any other resources please specify them. There are a lot of chemical classes which offer chemical diversity. Did the authors use a specific criterion to find the desired compounds?

33. I argue again that the pharmacophore step is not relevant and helpful. Indeed, it looks excellent as images and information as well as a method, but if not used as a query to find new candidates with improved properties is just an extra method and a lot of work. The filtration step (as the authors responded) could easily have been performed with a simple visual analysis of the 30 ligands or as already suggested by docking them.

44.  What was the reason for selecting the PDB ID: 5R84 target?

55.  Did the authors try (just for personal curiosity) to dock indinavir in the active site of the 5R84 target? The authors used indinavir and GWS in ADMET studies (e.g. new Table 3.) as references, but avoided to use indinavir in the docking step? The main goal is to find the most promising Mpro inhibitor and indinavir is already being tested/investigated as a candidate for SARS-CoV-2. Why did the authors choose indinavir instead of any other similarly antiviral drugs (Remdesivir, Dolutegravir, Raltegravir, Lopinavir, etc)?

66.  Figures 8, 9, and 10 should be merged into one figure and mentioned in the text at the appropriate place. 

Author Response

Thank you for your efforts and your valuable comments. All comments were considered in high interest and all changes were highlighted in the revised manuscript.

1- Figures 6 and 7 should be merged into one figure, although, in the referee's opinion, both figures could be moved to supplementary and keep only a brief explanation.

Response: Done

2- The authors added a short explanation about the selection of the 310 secondary metabolites. The explanation is not satisfactory. Where did the authors find these compounds? In literature or any other resources? If literature, please add the appropriate references. If any other resources, please specify them. There are a lot of chemical classes which offer chemical diversity. Did the authors use a specific criterion to find the desired compounds?

Response: Explained in detail

3- I argue again that the pharmacophore step is not relevant and helpful. Indeed, it looks excellent as images and information as well as a method, but if not used as a query to find new candidates with improved properties is just an extra method and a lot of work. The filtration step (as the authors responded) could easily have been performed with a simple visual analysis of the 30 ligands or as already suggested by docking them.

Response: Done

4- What was the reason for selecting the PDB ID: 5R84 target?

Response: Our team used all the crystal proteins of SARSCoV-2 in other studies. In the current work, we used 5R84 as a biological target and its co-crystallized ligand as a reference molecule.

5- Did the authors try (just for personal curiosity) to dock indinavir in the active site of the 5R84 target? The authors used indinavir and GWS in ADMET studies (e.g. new Table 3.) as references, but avoided to use indinavir in the docking step? The main goal is to find the most promising Mpro inhibitor and indinavir is already being tested/investigated as a candidate for SARS-CoV-2. Why did the authors choose indinavir instead of any other similarly antiviral drugs (Remdesivir, Dolutegravir, Raltegravir, Lopinavir, etc)?

Response: Thank you for these questions.

- First, we did not carry out the docking for indinavir because we used the co-crystallized ligand as a reference molecule and there was no need to select another reference molecule as indinavir. The selection of the co-crystallized ligand as a reference molecule because it has a standard, reported, and correct binding mode against the target receptor.

- We used indinavir as a reference molecule in the ADMET studies because it is an FDA-approved protease inhibitor drug and comparing the tested compound with an FDA-approved molecule gives strong results for the pharmacokinetic profile.

- We did not select Remdesivir, Dolutegravir, and Raltegravir as reference molecules because all of them act by different mechanisms

- Remdesivir: RNA-dependent RNA polymerase),

-Dolutegravir: HIV integrase inhibitor

- Raltegravir: HIV integrase inhibitor

- There was not any consideration for the exclusion of Lopinavir. We just select one drug (indinavir) from the family of protease inhibitors. In future work, we can use Lopinavir as a reference molecule.

6- Figures 8, 9, and 10 should be merged into one figure and mentioned in the text at the appropriate place.

Response: Done

Reviewer 2 Report

Dear authors,

Your manuscript has significantly improved.

Great amount of work and well written.

Author Response

Thank you for your praise and valuable revision.

Reviewer 3 Report

The authors have addressed and solved most of the deficiencies detected in the first revision. Only minor changes should be introduced in the manuscript before its publication:

1) The writing of the ADMET section must be improved. Parameters like "BBB" or acronyms like "CNS" must be defined before explaining ADMET results and the significance of Figure 11.

2) The description of the connection between the HOMO/LUMO topologies and the interactions observed in the protein-ligand complex is very similar to that presented in the MEP analysis. For this reason, I suggest combining the two descriptions in order to avoid text repetition.

Author Response

Thanks for your valuable comments. We have addressed all the points you requested.

Round 3

Reviewer 1 Report

In attention of the manuscript authors,

The authors satisfactorily responded to all referee’s requirements and made all the changes addressed in the manuscript. The manuscript has been substantially improved in both chemical content and English.

In this context, I agree that the manuscript be accepted for publication in the IJMS journal, in its present form.